Tracheostomy in children with congenital heart disease: a national analysis of the Kids’ Inpatient Database

Maxwell Bryan G. bryanmaxwell@gmail.com
McMillan Kristen Nelson
Department of Anesthesiology and Critical Care Medicine, School of Medicine, Johns Hopkins University , Baltimore, MD , USA
Zuo Li
Electronic publication date: 2014 Sep 11
Publication date: 2014
Volume: 2
Electronic Location ID: e568
Received 2014 Jun 20; Accepted 2014 Aug 19
Copyright: © 2014 Maxwell and McMillan
Copyright year: 2014
Copyright holder: Maxwell and McMillan
License: This is an open access article distributed under the terms of the Creative Commons Attribution License, which permits unrestricted use, distribution, reproduction and adaptation in any medium and for any purpose provided that it is properly attributed. For attribution, the original author(s), title, publication source (PeerJ) and either DOI or URL of the article must be cited.
License URL: https://creativecommons.org/licenses/by/4.0/

Keywords: Tracheostomy, Congenital heart disease, Pediatrics, Respiratory failure, Pediatric critical care, Single-ventricle physiology

Funding: Open Access Promotion Fund of the Johns Hopkins University Libraries Publication of this article was funded in part by the Open Access Promotion Fund of the Johns Hopkins University Libraries. The funder had no role in study design, data collection and analysis, decision to publish, or preparation of the manuscript.

==============================
Background. While single-institution studies reported the indications and outcomes of tracheostomy in children with congenital heart disease (CHD), no national analyses have been performed. We sought to examine the indications, performance, outcomes, and resource utilization of tracheostomy in children with CHD using a nationally representative database.

Methods. We identified all children undergoing tracheostomy in the Kids’ Inpatient Database 1997 through 2009, and we compared children with CHD to children without CHD. Within the CHD group, we compared children whose tracheostomy occurred in the same hospital admission as a cardiac operation to those whose tracheostomy occurred without a cardiac operation in the same admission.

Results. Tracheostomy was performed in n = 2,495 children with CHD, which represents 9.6% of all tracheostomies performed in children (n = 25,928), and 3.5% of all admissions for children with CHD (n = 355,460). Over the study period, there was an increasing trend in the proportion of all tracheostomies that were done in children with CHD (p < 0.0001) and an increasing trend in the proportion of admissions for children with CHD that involved a tracheostomy (p < 0.0001). The population of children with CHD undergoing tracheostomy differed markedly in baseline characteristics, outcomes, and resource utilization. Similarly, the subgroup of children whose tracheostomy was performed in the same admission as a cardiac operation differed significantly from those whose tracheostomy was not.

Conclusions. Tracheostomy is an increasingly common procedure in children with CHD despite being associated with significantly greater resource utilization and in-hospital mortality. The population of children with CHD who undergo tracheostomy differs markedly from that of children without CHD who undergo tracheostomy, and important differences are observed between children who undergo tracheostomy in the same admission as a cardiac surgical procedure and those who undergo tracheostomy in a nonsurgical admission, as well as between children with single-ventricle physiology and children with two-ventricle physiology.

Introduction

Congenital heart disease (CHD) has become more prevalent worldwide (Van der Linde et al., 2011). Surgical techniques to treat even the most complex structural lesions, along with improved intensive care unit (ICU) capabilities, have improved survival for children with CHD (Nieminen, Jokinen & Sairanen, 2001). But with greater long-term survival of children with complex CHD and other significant comorbid conditions, a fraction of children with CHD require tracheostomy for airway protection and/or ventilator dependence. Many children experience a prolonged and difficult recovery after cardiac operations, and even after the initial recovery, children with CHD have greater vulnerability to subsequent critical illness (Székely et al., 2006). For instance, even routine respiratory infections can be life threatening and require ventilator support in children with complex, palliated CHD.

Single-institution studies (Challapudi, Natarajan & Aggarwal; LoTempio & Shapiro; Hoskote et al., 2005; Rossi et al., 2009; Cotts et al., 2011; Costello et al., 2014) have reported indications and outcomes of tracheostomy in this population, but these studies have been small (sample size range: 4–59) and lack external validity because of substantial heterogeneity in institutional practices surrounding postoperative ventilator management and decision making about tracheostomy. These studies report different patterns of how commonly tracheostomy is used in this population and the timing of its performance. No national analyses have been performed. We used an established, nationally representative administrative database to examine the indications, performance, outcomes, and resource utilization of tracheostomy in children with CHD nationwide. This database provides a novel opportunity to explore broad patterns of tracheostomy in the pediatric CHD population across geography, different practice environments, and different institutional preferences.

Methods

This study was exempt from institutional review board review because it uses publicly available, de-identified data. Administrative records were extracted from discharge datasets for the years 1997–2009 from the Kids’ Inpatient Database (KID), part of the Healthcare Cost and Utilization Project (HCUP) of the Agency for Healthcare Research and Quality. KID is the largest publicly available, all-payer database of inpatient pediatric care in the United States. Each dataset, released every third year, includes records on a sample of discharges from all community, non-federal, non-rehabilitation hospitals in states that participate in HCUP. Hospitals are identified in KID by type according to classifications established by the National Association of Children’s Hospitals and Related Institutions (NACHRI): children’s general hospitals, children’s specialty hospitals, children’s units in a general hospital, or not identified as a children’s hospital.

KID contains discharge sample weights to facilitate nationally representative estimates based on the sampling design; these weights adjust for growth in participation from 1997, when 2,521 hospitals in 22 participating states submitted records reflective of 6.7 million national discharges, to 2009, when 4,121 hospitals in 44 states submitted records reflective of 7.4 million national discharges. While KID contains limited data on each inpatient encounter, its size and sampling frame facilitate the analysis of comparatively rare clinical events at a national level.

Diagnostic, comorbidity, and procedural information was based on HCUP-supplied international classification of diseases, ninth revision, clinical modification (ICD-9-CM) codes: tracheostomy (Volume 3 procedure codes 31.1-2), congenital heart disease (745.x, 746.x, and 747.1-4), single-ventricle anatomy (747.7, 745.3, and procedure code 39.21), pulmonary hypertension (416.x), genetic syndromes (758.0-5), vocal cord paralysis/paresis (478.30-34), tracheal or bronchial pathology (519.1, 519.19 and 748.3) and respiratory failure (518.5x, 518.8x, 770.84, 799.1, and V46.1-2). Cardiac surgical procedure codes were used to define whether the tracheostomy was performed during an admission that included a cardiac operation or an admission that did not. A composite comorbidity point score was calculated based on the van Walraven modification of the Elixhauser comorbidity measure (Elixhauser et al., 1998).

We compared children with CHD who underwent tracheostomy (“CHD group”) to children without CHD who underwent tracheostomy (“non-CHD group”). Within the CHD group, we compared admissions of children whose tracheostomy occurred in the same hospital admissions as a cardiac operation (“cardiac surgical group”) to those whose tracheostomy occurred without a cardiac operation in the same admission (“nonsurgical group”). Because patients with single-ventricle physiology are a subgroup of particular interest, we also compared children with single-ventricle physiology to those with two-ventricle physiology.

Discharge weights were used to create national estimates within the KID sampling frame. Primary outcome measures included inpatient length of stay (variable LOS), timing of the performance of tracheostomy relative to admission and cardiac surgery (calculated from variables PRDAYx), total inpatient hospital charges (variable TOTCHG), and in-hospital mortality (variable DIED). Hospital charges were indexed to inflation by adjusting all values to 2009 dollars using the Bureau of Labor Statistics Consumer Price Index subindex specific to inpatient hospital services with a baseline of December 1996 taken as 100 (Bureau of Labor Statistics).

Statistical analysis

Because the sampling frame of the KID requires the use of advanced techniques (facilitated by PROC SURVEYMEANS in SAS) to estimate variance, continuous variables are presented as mean ± standard error. Discrete variables are presented as number (percentage). Inter-group comparisons were carried out using the Mann–Whitney–Wilcoxon test for continuous variables and Fisher’s exact test or Pearson’s chi-squared test for categorical variables, as appropriate. Trends over time were examined by month of hospital admission using a seasonal Mann–Kendall test for trend (a nonparametric test to determine the presence and direction of a trend over time) (Hirsch, Slack & Smith, 1982) and accounts for seasonal variability, since inpatient pediatric care might involve seasonal fluctuations due to, for instance, patterns of winter illness, surgical scheduling in relation to school calendars, etc.). For trend analyses only, records with missing data for the month of admission (8.6% of all KID records) were excluded; these records were included for all non-trend analyses and descriptive data. A predetermined alpha of 0.05 was used as the threshold of statistical significance. Trend analyses were performed using R (R 3.0.3; The R Foundation for Statistical Computing, Vienna, Austria). All other analyses were performed using SAS (SAS 9.3; SAS Institute, Cary, NC, USA).

Results

Tracheostomy was performed in n = 2,495 children with CHD, which represents 9.6% of all tracheostomies performed in children (n = 25,928), and 3.5% of all admissions for children with CHD (n = 355,460).

The performance of tracheostomy demonstrated a significantly increasing trend over time among children with and without CHD (p < 0.0001 for both, Fig. 1). Over the study period, children with CHD accounted for a significantly increasing proportion of all tracheostomies (p < 0.0001; Fig. 2), and tracheostomy admissions accounted for an increasing proportion of admissions among children with CHD (p < 0.0001, Fig. 3).

Figure 1 National estimates of total admissions including the performance of tracheostomy by month.

Children with CHD (blue bars; increasing trend, p < 0.0001); children without CHD (red bars; increasing trend, p < 0.0001).

Figure 2 Proportion of all pediatric admissions with performance of a tracheostomy accounted for by children with CHD, by month.

Increasing trend, p < 0.0001.

Figure 3 Percentage of all admissions of children with CHD that involve performance of a tracheostomy.

Increasing trend, p < 0.0001.

Table 1 shows baseline characteristics for the CHD and non-CHD groups. The CHD group was predominantly (86%) neonates or infants, whereas the majority (63%) of the non-CHD group was older than age 10. The CHD group had a roughly even balance of boys and girls, whereas the non-CHD group was 64% male. A greater fraction of the CHD group received Medicaid, received care in urban teaching hospitals, and received care in a recognized children’s hospital or unit. Table 2 shows outcomes of tracheostomy admissions in the CHD and non-CHD groups. The CHD group underwent tracheostomy significantly later in their hospitalization, was less likely to be discharged to a skilled nursing or long-term care facility, and had a longer length of stay, greater total hospital charges, and greater in-hospital mortality.

Table 1 Characteristics of children with and without congenital heart disease undergoing tracheostomy.

	CHD	No CHD	p	
	n = 2,495	n = 23,433		
	n	%	n	%		
Age (years)	1.2	±0.1	11.2	±0.2	<0.0001	
Age category					<0.0001	
Birth admission	513	(20.6%)	1,453	(6.2%)		
<1 yr, readmission since birth	1,631	(65.4%)	4,462	(19.0%)		
Age 1–4	190	(7.6%)	1,771	(7.6%)		
Age 5–9	42	(1.7%)	940	(4.0%)		
Age ≥ 10	119	(4.8%)	14,807	(63.2%)		
Sex					<0.0001	
Male	1,284	(51.5%)	15,083	(64.4%)		
Female	1,211	(48.5%)	8,350	(35.6%)		
Race					<0.0001	
White	913	(36.6%)	9,682	(41.3%)		
Black	391	(15.7%)	3,803	(16.2%)		
Hispanic	465	(18.6%)	3,311	(14.1%)		
Asian	51	(2.0%)	475	(2.0%)		
Other/Missing	675	(27.0%)	6,162	(26.3%)		
Van Walraven score	4.0	±0.2	4.1	±0.1	0.2958	
Genetic syndrome	254	(10.2%)	353	(1.5%)	<0.0001	
Pulmonary hypertension	241	(9.7%)	405	(1.7%)	<0.0001	
Vocal cord paralysis/paresis	150	(6.0%)	864	(3.7%)	<0.0001	
Tracheal/bronchial pathology	706	(28.3%)	2,428	(10.4%)	<0.0001	
Respiratory failure	1,383	(55.4%)	1,374	(58.6%)	0.008	
Payer					<0.0001	
Medicare	b	b	84	(0.4%)		
Medicaid	1,390	(55.7%)	9,936	(42.4%)		
Private insurance	947	(38.0%)	10,859	(46.3%)		
Self-pay/other	154	(6.2%)	2,554	(10.9%)		
Hospital setting					<0.0001	
Rural	16	(0.7%)	415	(1.8%)		
Urban, nonteaching	250	(10.0%)	3,501	(14.9%)		
Urban, teaching	2,229	(89.3%)	19,517	(83.3%)		
Hospital type a					<0.0001	
Children’s general hospital	759	(30.4%)	3,393	(14.5%)		
Children’s specialty hospital	b	b	42	(0.2%)		
Children’s unit in a general hospital	1,103	(44.2%)	9,085	(38.8%)		
Not identified as a children’s hospital	633	(25.4%)	10,913	(46.6%)		
Notes.

a As defined by the National Association of Children’s Hospitals and Related Institutions.

b Number not reported in keeping with KID privacy rules for cells with n < 10. True values used to calculate p values.

Values are number (percentage) or mean ± standard error, as appropriate.

Table 2 Outcomes of tracheostomy in children with and without congenital heart disease.

	CHD	No CHD	p	
	n = 2, 495	n = 23,433		
	n	%	n	%		
Days from admission to tracheostomy	47.0	±1.8	18.4	±0.5	<0.0001	
Length of stay (days)	98.6	±2.5	47.4	±0.9	<0.0001	
Total hospital charges	$603,651	±$16,479	$368,956	±$7,344	<0.0001	
Discharge disposition (of alive discharges)					<0.0001	
Home, routine	862	(41.3%)	7,673	(35.1%)		
Home with home health	581	(27.8%)	3,571	(16.3%)		
Nursing facility/subacute/long-term care	645	(30.9%)	10,636	(48.6%)		
Mortality	407	(16.3%)	1,553	(6.6%)	<0.0001	
Notes.

Values are number (percentage) or mean ± standard error, as appropriate.

Table 3 shows baseline characteristics for the CHD group, divided into cardiac surgical and nonsurgical subgroups. Children in the cardiac surgical group were less likely to have tracheal or bronchial pathology and more likely to have respiratory failure as an indication for tracheostomy. Table 4 shows outcomes of tracheostomy admissions in the cardiac surgical and nonsurgical groups. The majority (86%) of CHD patients in the cardiac surgical group underwent tracheostomy after their cardiac operation, an average of 49.6 ± 3.1 days after the cardiac operation and 72.3 ± 4.5 days after initial hospital admission.

Table 3 Characteristics of children with congenital heart disease undergoing tracheostomy in admissions involving cardiac surgery or nonsurgical admissions.

	Cardiac surgery	No cardiac surgery	p	
	n = 590	n = 1,905		
	n	%	n	%		
Age (years)	1.3	±0.2	1.1	±0.1	0.11	
Age category					<0.0001	
Birth admission	64	(10.9%)	448	(23.5%)		
<1 yr, readmission since birth	431	(73.0%)	1,200	(63.0%)		
age 1–4	56	(9.5%)	134	(7.0%)		
age 5–9	b	b	36	(1.9%)		
age ≥10	32	(5.5%)	87	(4.6%)		
Sex					0.76	
Male	300	(50.9%)	984	(51.7%)		
Female	290	(49.1%)	921	(48.3%)		
Race					0.0011	
White	221	(37.5%)	692	(36.3%)		
Black	53	(8.9%)	338	(17.8%)		
Hispanic	122	(20.8%)	343	(18.0%)		
Asian	b	b	43	(2.2%)		
Other/Missing	185	(31.4%)	489	(25.7%)		
Van Walraven score	5.2	±0.3	3.7	±0.2	<0.0001	
Genetic syndrome	51	(8.6%)	212	(11.1%)	0.20	
Pulmonary hypertension	68	(11.5%)	174	(9.1%)	0.18	
Vocal cord paralysis/paresis	54	(9.1%)	96	(5.1%)	0.016	
Tracheal/bronchial pathology	112	(18.9%)	488	(25.6%)	0.007	
Respiratory failure	400	(67.8%)	983	(51.6%)	<0.0001	
Payer					0.59	
Medicare	b	b	b	b		
Medicaid	327	(55.5%)	1,063	(55.8%)		
Private insurance	219	(37.1%)	729	(38.3%)		
Self-pay/other	44	(7.5%)	110	(5.8%)		
Hospital setting					0.031	
Rural	b	b	13	(0.7%)		
Urban, nonteaching	36	(6.2%)	213	(11.2%)		
Urban, teaching	550	(93.2%)	1679	(88.1%)		
Hospital type a					<0.0001	
Children’s general hospital	242	(41.1%)	516	(27.1%)		
Children’s specialty hospital	b	b	b	b		
Children’s unit in a general hospital	261	(44.3%)	842	(44.2%)		
Not identified as a children’s hospital	87	(14.7%)	547	(28.7%)		
Notes.

a As defined by the National Association of Children’s Hospitals and Related Institutions.

b Number not reported in keeping with KID privacy rules for cells with n < 10. True values used to calculate p values.

Values are number (percentage) or mean ± standard error, as appropriate.

Table 4 Outcomes of tracheostomy in children with congenital heart disease during cardiac surgical or nonsurgical admissions.

	Cardiac surgery	No cardiac surgery	p	
	n = 590	n = 1,905		
	n	%	n	%		
Days from admission to cardiac surgery	28.5	±4.0				
Days from admission to tracheostomy	64.8	±3.7	42.2	±1.9	<0.0001	
Timing of tracheostomy relative to cardiac surgery						
Same day	18	(3.1%)				
Tracheostomy before cardiac surgery	62	(10.5%)				
Days from admission to tracheostomy	44.3	±4.1				
Days from tracheostomy to cardiac surgery	46.6	±9.5				
Days from admission to cardiac surgery	74.1	±14.5				
Tracheostomy after cardiac surgery	510	(86.4%)				
Days from admission to cardiac surgery	22.7	±4.1				
Days from cardiac surgery to tracheostomy	49.6	±3.1				
Days from admission to tracheostomy	72.3	±4.5				
Length of stay (days)	119.2	±5.2	92.3	±2.5	<0.0001	
Total hospital charges	$844,914	±$32,124	$548,515	±$17,660	<0.0001	
Discharge disposition (of alive discharges)					0.20	
Home, routine	255	(43.2%)	776	(40.8%)		
Home with home health	139	(23.5%)	553	(29.1%)		
Nursing facility/subacute/long-term care	196	(33.2%)	576	(30.3%)		
Mortality	133	(22.5%)	274	(14.4%)	<0.0001	
Notes.

Values are number (percentage) or mean ± standard error, as appropriate.

Table 5 shows outcomes of tracheostomy admissions in children with single-ventricle physiology (n = 169, 6.8% of all CHD tracheostomy admissions) compared to children with two-ventricle physiology. Compared to two-ventricle CHD patients, single-ventricle patients had a higher average age (1.9 ± 0.4 vs. 1.1 ± 0.1 years), underwent tracheostomy later in their admission (60.0 ± 5.0 vs. 46.2 ± 1.9 days), had greater total hospital charges ($754,764 ± 47,887 vs. $593,437 ± $17,028), and much greater in-hospital mortality (35.5% vs. 14.9%).

Table 5 Outcomes of tracheostomy in children with congenital heart disease and single-ventricle physiology compared to children with congenital heart disease and two-ventricle physiology.

	Single-ventricle	Two-ventricle	p	
	n = 169	n = 2,326		
	n	%	n	%		
Age (years)	1.9	±0.4	1.1	±0.1	0.003	
Days from admission to tracheostomy	60.0	±5.0	46.2	±1.9	0.0005	
Cardiac surgical admission	110	(65.1%)	480	(20.6%)	<0.0001	
Days from admission to cardiac surgery	21.6	±9.2	29.8	±4.4	0.22	
Timing of tracheostomy relative to cardiac surgery					0.11	
Same day			13	(2.7%)		
Tracheostomy before cardiac surgery			56	(11.7%)		
Tracheostomy after cardiac surgery	99	(90.0%)	411	(85.6%)		
Length of stay (days)	104.2	±7.4	98.2	±2.6	0.029	
Total hospital charges	$754,764	±$47,887	$593,437	±$17,028	<0.0001	
Discharge disposition (of alive discharges)					0.36	
Home, routine	74	(43.8%)	957	(41.1%)		
Home with home health	34	(20.1%)	656	(28.2%)		
Nursing facility/subacute/long-term care	61	(36.1%)	713	(30.7%)		
Mortality	60	(35.5%)	347	(14.9%)	<0.0001	
Notes.

a Number not reported in keeping with KID privacy rules for cells with n < 10. True values used to calculate p values.

Values are number (percentage) or mean ± standard error, as appropriate.

Discussion

The principal finding of this study is that tracheostomy is an increasingly common procedure in children with CHD despite being associated with significantly greater resource utilization and in-hospital mortality.

In both the CHD and non-CHD groups, we observed an increasing trend in the proportion of admissions involving tracheostomy. This may reflect a shift in practice towards the belief that earlier tracheostomy improves outcomes for critically ill patients, which has been well-established in adults (Gomes Silva et al., 2012) and has begun to gain acceptance in other pediatric populations, including trauma (Holscher et al., 2014) and burns (Palmieri, Jackson & Greenhalgh, 2002). However, even beyond this broader overall increase in tracheostomy, we found that children with CHD accounted for an increasing proportion of all tracheostomy admissions. This may reflect increasing acceptance of tracheostomy by congenital heart surgeons as a bridge to recovery in critically ill patients, increasing complexity of cases (with a greater attendant risk of prolonged ICU recovery), or expansion of surgical programs to include children with other comorbid conditions (e.g., pulmonary hypertension) that might previously have been viewed as contraindications to surgical candidacy and that complicate their recovery from congenital heart surgery.

The population of children with CHD who undergo tracheostomy differs markedly from that of children without CHD who undergo tracheostomy. The CHD group mostly was composed of a gender-balanced population of neonates and infants at urban, tertiary, children’s hospitals, whereas the non-CHD group was comprised mostly of pre-teen and teenage boys. These observations suggest a bimodal pattern of pediatric critical illness, in which CHD is a major contributor in the neonatal and infant period, but a comparatively less common contributor in the school-age and adolescent age group.

Within the population of children with CHD undergoing tracheostomy, we observed important differences between those whose tracheostomy occurred in the same hospital admissions as a cardiac operation and those whose tracheostomy did not. The cardiac surgical group had higher baseline comorbidity scores, were more likely to have care in urban teaching hospitals and in an identified children’s hospital or unit. The cardiac surgical group was also less likely to have tracheal or bronchial pathology and more likely to have respiratory failure as an indication for tracheostomy.

The cardiac surgical group underwent tracheostomy later in their hospitalization, had a longer length of stay, greater total hospital charges, and greater in-hospital mortality. Within the group that had both a tracheostomy and a cardiac surgical procedure in the same admission, the timing of the two procedures was notable. Most children (86.4%) had cardiac surgery prior to their tracheostomy; a minority (10.5%) had a tracheostomy prior to cardiac surgery. Performance of tracheostomy on the same day as a cardiac operation was rare (3.1%).

Children with single-ventricle physiology undergoing tracheostomy were older than those with two-ventricle CHD; the mean age of 1.9 years in the single-ventricle group suggests a predominance of tracheostomies occurring remote from initial palliation, though longitudinal information on sequential surgical procedures is not available. The single-ventricle group was more likely to reflect a tracheostomy performed in a surgical admission, and the longer interval to tracheostomy suggests a prolonged ICU course in an attempt to avoid tracheostomy. This group demonstrated high mortality, with more than 1 in 3 children in the single-ventricle group dying in the same admission in which tracheostomy is performed. We suspect that in this context, tracheostomy is a marker for complex critical illness, but it is also important to note that a causal role may exist, as prolonged positive pressure ventilation may have significant pathophysiologic effects on the single-ventricle circulation (Shekerdemian et al., 1997). Further investigations would be needed to determine the benefits and risks of early compared to late tracheostomy in single-ventricle patients with respiratory failure.

Our results are consistent with the prior single-institution analyses that have demonstrated significant mortality and long length of stays in pediatric CHD patients who require tracheostomy. Our analysis provides a comprehensive, national depiction of the timing of tracheostomy in cardiac surgical and nonsurgical admissions; prior analyses have provided widely varying information about timing. Our analysis also adds important information about trends over time and across the entire nation, information on resource utilization associated with tracheostomy admissions, and a benchmark to which future in-depth, single-institution studies can compare their practices.

We believe it is important to note that KID is a database of admissions, so this analysis does not provide longitudinal data on patients with a tracheostomy through multiple hospital admissions or outpatient events. Any individual child who has multiple hospitalizations may appear in KID multiple times; no mechanism allows for connecting the records from separate admissions, and one must interpret this analysis in light of this feature. For instance, we cannot determine the timing or frequency of readmissions in patients with tracheostomy after the initial admission when the tracheostomy was performed. Administrative data also lack clinical details, including ventilator dependence, tracheostomy decannulation, and tracheostomy-associated surgical complications, so this analysis cannot add information about those dimensions of tracheostomy performance in children with CHD.

Several limitations to this study exist as a result of structural features of the KID database. First, all administrative analyses are limited by classification error. However, such error is less likely to occur with respect to procedural codes that are important to reimbursement; hospitals have an incentive to ensure that all performed procedures are coded properly. Similarly, important comorbidity information (e.g., CHD) is more likely to be reliably coded than minor coexisting conditions of less significance to the primary problem treated in the admission. These features are likely to mitigate the effects of classification error in our analysis of a major, conspicuous procedure in a population with a major cardiac condition.

Classification error is likely to be a particular concern in our analysis of single-ventricle patients, as the diagnosis codes we used to identify single-ventricle physiology are incomplete. Some diagnoses (for instance, atrioventricular canal) include a heterogeneous range of defects that may involve either single-ventricle or two-ventricle physiology, depending on severity. ICD-9 codes cannot differentiate the single-ventricle subset of these patients. However, it is most likely that classification error underestimates the single-ventricle population, suggesting that the differences we observed might be even more pronounced if we were able to identify all single-ventricle patients.

Second, resource utilization information is available only for the entire hospital admission, which precludes knowledge of the distribution of individual contributors to resource utilization (e.g., medication costs, supplies, procedures). This analysis offers information on resource utilization in admissions of patients undergoing tracheostomy, but it cannot elucidate the resources specifically devoted to tracheostomy itself as compared to other care provided in the same admission.

Third, total hospital charges as reported in the NIS represent hospital billing, not actual expenditures or reimbursement. The complex, nonlinear relationships between hospital charges, hospital costs, insurance reimbursement, and patient co-payments complicate the extrapolation of these results to an estimation of the true societal cost of inpatient care in children with a tracheostomy. However, to the degree that the relationship between charges, costs, and other parameters does not change drastically over short periods of time, hospital charges at minimum are useful for analysis of resource utilization trends, allow for aggregation of data at the national level, and give some sense of the magnitude of resource utilization devoted to this population.

Fourth, KID provides data only triennially, and its sampling frame does change slightly from year to year. Discharge weights adjust to produce national estimates for each year, and we also attempted to address this by ensuring that the ICD9 codes used to identify tracheostomy have not changed during the study period (National Center for Health Statistics and the Centers for Medicare & Services, 2013).

Fifth, the trend analyses were limited by a small amount of missing data (8.6% of all records did not include the month of admission, only the year). We believe the benefit of greater data granularity outweighed the downsides of this limitation, as use of year only would significantly limit the power of trend analyses and would prohibit any adjustment for seasonal fluctuations.

Despite these limitations, this analysis is useful in demonstrating the relative trends in performance of tracheostomy and its resource utilization implications. It captures national practice patterns and outcomes to demonstrate that tracheostomy is an increasingly common intervention in children with CHD, and in children undergoing heart surgery during the same admission, is usually performed significantly later (on average, 49 days) in the admission. It outlines important demographic and clinical differences between children with CHD who undergo tracheostomy and children without CHD who undergo tracheostomy. Future studies must build on these results to further investigate the benefits of early vs. late tracheostomy to determine if the significant resource utilization associated with tracheostomy improves overall resource utilization, recovery, or long-term outcomes. Our subgroup analysis of single-ventricle patients suggests that in particular, focused studies analyzing the benefits of tracheostomy in this subpopulation will be revealing.

Additional Information and Declarations

Competing Interests

Author Contributions

Human Ethics

Data Deposition

The authors declare there are no competing interests.

Bryan G. Maxwell conceived and designed the experiments, performed the experiments, analyzed the data, wrote the paper, prepared figures and/or tables, reviewed drafts of the paper.

Kristen Nelson McMillan conceived and designed the experiments, analyzed the data, wrote the paper, reviewed drafts of the paper.

The following information was supplied relating to ethical approvals (i.e., approving body and any reference numbers):

This study was exempt from institutional review board review because it uses publicly available, de-identified data.

The following information was supplied regarding the deposition of related data:

The data is available at the HCUP: http://www.hcup-us.ahrq.gov/kidoverview.jsp.

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
