# Peer review of "Tracheostomy in children with congenital heart disease: a national analysis of the Kids’ Inpatient Database"

_PeerJ, doi:10.7717/peerj.568_

## Round 0.1 · original submission · Major Revisions

Please fully address the reviewers' comments.

Reviewer 1 ·

Basic reporting

The result section appears abbreviated. I would have preferred the authors have a more granular discussion of the results in the actual result section. Perhaps more discussion of the different associations of the non CDH patients. There is an extensive review of a large database.

Experimental design

No comments

Validity of the findings

There was extensive discussion of the limitations of their data collection and the methodology which gives the impression of undermining the overall conclusions made by the authors. The authors do not address any etiologies of the overall increase in incidence of tracheostomies, particularly in the nonCHD patients.

Additional comments

I would recommend that the authors expand the result section and perhaps put the numerous limitations of the database in slightly better context.

Reviewer 2 ·

Basic reporting

1. Please specify the significance of this study in the introduction part.

2. Please have a more detailed result analysis and explanation in the result part.

Experimental design

1. Is that possible to include a study group that has CHD but without tracheostomy?
2. It might be not appropriate to compare the CHD and non-CHD group,since those patients themselves are having different health background.

Validity of the findings

No comments

Additional comments

1. Please specify the significance of this study in the introduction part.
2. Please have a more detailed result analysis and explanation in the result part.
3. Is that possible to include a study group that has CHD but without tracheostomy?
4. It might be not appropriate to compare the CHD and non-CHD group,since those patients themselves are having different health background.

Reviewer 3 ·

Basic reporting

this work provides a national analyses of the indications and outcomes of tracheostomy in children with CHD and without CHD using a nationally representative database. It provides novel knowledge in the field of tracheostomy.

Experimental design

No comments

Validity of the findings

No Comments

Additional comments

The introduction is rather limited. It would be helpful to expand.
The Results should include more detailed discussion about the Tables.
Some part of the discussion should be included in the Results, such as the second paragraph.

---

## Round 0.2 · Minor Revisions

Please address the reviewers' comments.

Reviewer 2 ·

Basic reporting

No Comments

Experimental design

No Comments

Validity of the findings

No comments

Additional comments

The authors may want to pay attention to the format of those tables, especially in table 3, the right parenthesis should be in the same line as the number instead of in the second line.

Reviewer 4 ·

Basic reporting

1. The current manuscript title is too general, which does not reflect the limitations of structural features of the current database, the more specific ones maybe better.

2. The information from the Introduction part is limited, which should include more background about the performance of tracheostomy in children with and without CHD, same as the cardiac operation. And you may also mention the significance of the current analysis based on the whole US database.

3. In Discussion, you would better to compare the difference in conclusions between previous single-institution studies and your national analysis to further demonstrate the advantages of necessary and sufficient statistic in the relative trends analyses.

Experimental design

No Comments

Validity of the findings

No Comments

Additional comments

I would recommend that the authors should present it more clearly and carefully for the conclusions based on the limited database and potential implications for future peer-studies in Abstract and Results parts.

---

## Round 0.3 · Minor Revisions

Thank you for the revision. However, I would like you to add a reference for the "Many children experience a prolonged .....CHD". In additon, the reviewer suggested that "I would recommend that the authors should present it more clearly and carefully for the conclusions based on the limited database and potential implications for future peer-studies in Abstract and Results parts." This means, it would be very important to address all of these concerns: Amongst these concerns are: the need for better SYNTHESIS. It would be essential to group together sets of related data and synthesize what the reports mean in the broader context. What - together - do the related sets of observations tell us, how do they further our understanding, and what "holes" or gaps in our knowledge remain that need to be filled?

---

## Round 0.4 · accepted · Accept

Thank you for the revision. Please note you should fix more typos in the reference and text during the proof stage since there are still some. For example, "have improved survival for children with CHD.(Nieminen, Jokinen & Sairanen, 2001) But.... "-- missing the period before" but".